# The Ultratrace Determination of Fluoroquinolones in River Water Samples by an Online Solid-Phase Extraction Method Using a Molecularly Imprinted Polymer as a Selective Sorbent

**DOI:** 10.3390/molecules27238120

**Published:** 2022-11-22

**Authors:** A. N. Baeza, Idoia Urriza-Arsuaga, F. Navarro-Villoslada, Javier L. Urraca

**Affiliations:** 1Institute of Science and Technology of Materials, University of Havana, Zapata y G, La Habana 10400, Cuba; 2Independent Researcher, 28007 Madrid, Spain; 3Department of Analytical Chemistry, Faculty of Chemistry, Complutense University of Madrid, Plaza Ciencias, 2, 28040 Madrid, Spain

**Keywords:** molecularly imprinted polymers, fluoroquinolones, solid phase extraction, online SPE, water samples

## Abstract

Fluoroquinolones (FQs) are broad-spectrum antibiotics widely used to treat animal and human infections. The use of FQs in these activities has increased the presence of antibiotics in wastewater and food, triggering antimicrobial resistance, which has severe consequences for human health. The detection of antibiotics residues in water and food samples has attracted much attention. Herein, we report the development of a highly sensitive online solid-phase extraction methodology based on a selective molecularly imprinted polymer (MIP) and fluorescent detection (HPLC-FLD) for the determination of FQs in water at low ng L^−1^ level concentration. Under the optimal conditions, good linearity was obtained ranging from 0.7 to 666 ng L^−1^ for 7 FQs, achieving limits of detection (LOD) in the low ng L^−1^ level and excellent precision. Recoveries ranged between 54 and 118% (RSD < 17%) for all the FQs tested. The method was applied to determining FQs in river water. These results demonstrated that the developed method is highly sensitive and selective.

## 1. Introduction

The increase in antibiotic resistance has become a serious problem worldwide [1]. The presence of resistant bacteria in hospitals and communities has been associated with the indiscriminate use of antimicrobials during treatments, remaining one of the fundamental problems of modernity [2]. With the advances in science and access to antibiotics for the world population, new challenges have arisen regarding their use in humans and animals to allow higher production [3]. The indiscriminate use of these compounds has increased the presence of antibiotics in wastewater [4] and in foods of animal origin [5], in which due to their low concentrations, they lose their potency, allowing the bacteria to become resistant, and incapable of being killed.

When it comes to environmental pollution, the occurrence of these compounds in different bodies of water has been manifested in various parts of the world [6,7]. Among the compounds of greatest interest are fluoroquinolones (FQs), which, with their stability and low biodegradability, result in long persistence in the aquatic environment [8,9]. Today, there is considerable interest in studying and determining FQs in the environment, being a problem identified in both developed and underdeveloped countries [10,11]. In Cuba, not many studies have been carried out aimed at understanding the origin of these compounds in waters and their relationship with the presence of resistance genes [12]. Unfortunately, only a very few studies reported the presence of FQs in surface waters, especially in Havana City (Cuba), for instance the studies of the Quibú river [13]. Usually, the dilution that the compounds undergo in the environment and the access to methodologies with low limits of detection (LOD) limit the monitoring of environmental waters.

A wide variety of procedures and techniques have been applied for the extraction and quantification of FQs in surface water samples. In this sense, the online extraction methodologies [11], liquid-liquid extraction with subsequent salting-out [14], solid phase extraction (SPE) by mixed-bed sorbents, C18, or OASIS HLB [4,15], capillary electrophoresis [16], magnetic mixed hemimicelles [17], restricted access materials [18], boron-rich monoliths [19], Kraft’s lignin [20], and molecularly imprinted polymers (MIP) can be mentioned [21,22,23,24].

The preferred analytical methods involve liquid chromatography (HPLC) analysis combined with mass detection [11,14,15], fluorescence (FLD) [4,13], and ultraviolet detection [17,19]. Flow injection analysis with chemiluminescent detection [13] and electrochemical measurements [16] have also been reported.

In online SPE procedures, preconcentration and transfer of the whole sample to the chromatographic system translate in improved limits of detection. Therefore, the analytical method developed in the present work allows the ultra-trace detection of fluoroquinolones employing an adsorbent material (MIP) highly selective to these analytes for their extraction and clean-up. Moreover, the adsorbent material can be reused because the volume sample used in the analysis is higher than typical methods, minimizing in this way errors in the determination of the target molecules and the cost per sample can also be reduced. Automation and minimal or no sample manipulation are other important features of the proposed method. Finally, the online system used was affordable and easy to implement, allowing it to be fully automated and work as an early-warning or on-site monitoring system.

In this work, we describe the use of a molecularly imprinted polymer (MIP) as a selective sorbent for an automated online SPE-HPLC-FLD extraction and the determination of FQs at the low ng L^−1^ level and its application to river water samples.

## 2. Materials and Methods

### 2.1. Chemicals

Antibiotics (Figure 1) enrofloxacin (ENRO), norfloxacin (NOR), lomefloxacin (LOME), enoxacin (ENOX), levoxacin (LEVO), ciprofloxacin (CIPRO), methacrylic acid (MAA), 2-trifluoromethacrylic acid (TFMAA) and 2-hydroxyethyl ethylene glycol dimethacrylate (EDMA) were obtained from Sigma-Aldrich (St. Louis, MO, USA). 2,2′-azobis(2,4-dimethylvaleronitrile) (ABDV) was purchased from Wako (Neuss, Germany) and used as received. Sarafloxacin hydrochloride (SARA), was a gift from Fort Dodge veterinaria (Girona, Spain). Danofloxacin (DANO) was purchased from Riedel-de-Haën (Seelze, Germany).

Acetonitrile (ACN) and methanol (MeOH) (HPLC-grade) were provided by SDS (Peypin, France) and trifluoroacetic acid (TFA) (HPLC-grade, 99%) was from Fluka (Buchs, Switzerland).

Water was purified using a Milli-Q system (Millipore, Bedford, MA, USA). The monomers were purified, as required, by chromatography immediately before use, using an inhibitor–remover from Aldrich (Milwaukee, WI, USA). All solutions prepared for the HPLC were passed through a 0.45 µm nylon filter before use. 2-[4-(2-hydroxyethyl)-1-piperazinyl]ethanesulfonic acid (HEPES) was supplied by Aldrich (Steinheim, Germany). Trifluoroacetic acid (TFA) (HPLC-grade) was from Fluka (Buchs, Switzerland).

### 2.2. Polymer Synthesis

MIP/NIP particles were synthesized according to previous work [25]. Briefly, a pre-polymerization mixture was obtained by mixing 0.5 mmol of the template (ENOX) and 1 mmol of the functional monomers (MAA, TFMAA) solved in 1 mL of ACN, mixed with 20 mmol of the crosslinker (EDMA) and 2% (weight of the final mixture) of the initiator (ABDV). Then 7 g of silica (Si-500 40–75 µm, Silicycle) were placed in a 100 mL glass vial and mixed by stirring with 3.3 mL of the cocktail solution until the silica beads were freely flowing. Then the vial was sealed with a septum and the system was purged with N2 for 5 min. The polymerization was carried out in an oven at 60 °C for 24 h.

After polymerization, etching was performed by adding 3 × 140 mL of an aqueous solution of ammonium hydrogen difluoride (3 M) and shaking for 24 h. The final solid obtained was washed with water (until pH ~ 7), methanol/TFA 99/1 (1 L) and methanol (0.5 L). Finally, the solid was dried in vacuum at 50 °C for 24 h. Before use, they were sedimented using MeOH/water (80/20, *v*/*v*) to remove fine particles. NIP polymer was prepared in the same way but in the absence of template molecule.

### 2.3. Apparatus and Analytical Conditions

The pH of the buffer solutions and samples was adjusted with an ORION 710A pH/ISE meter (Beverly, MA, USA).

Chromatographic analysis was carried out with an HP-1200 HPLC from Agilent Technologies (Palo Alto, CA, USA) equipped with two quaternary pumps, online degasser, autosampler, automatic injector, column thermostat and a fluorescence (FLD) detector. The scheme of the configuration used for the online MISPE procedure is represented in Figure 2. The MISPE preconcentration column was connected to the HPLC system using a Rheodyne valve 7750E. Chromatographic separation was carried out on phenyl-hexyl analytical column (250 × 4.6 mm i.d., 3 µm) from Phenomenex (Torrance, CA, USA). The analytical separation was performed using a gradient elution combining solvent A (TFA 0.5%; pH 2.0), solvent B (acetonitrile) and solvent C (methanol) according to Appendix A. The initial conditions were maintained for 10 min at 1 mL min^−1^. The column temperature was kept at 30 °C. The fluorescence excitation/emission wavelengths were programmed at 280/440 nm for NOR, CIPRO, DANO, LOME, ENRO and SARA, and at 280/515 nm for LEVO.

Quantification was performed using external calibration peak area measurements. Linear calibration graphs were obtained in the following ranges: 3.3–333 µg L^−1^ for NOR and ENRO; 6.6–333 µg L^−1^ for LEVO, PRO and SARA; 6.6–666 µg L^−1^ for LOME, and 0.7–133 µg L^−1^ for DANO.

The preconcentration column consisted of a stainless-steel column (50 × 4.6 mm i.d.) packed with the NIP/MIP slurries in methanol, using MeOH/water (80/20, *v*/*v*) as the pushing solvent. The conditions used for the preconcentration, washing and elution of the samples are summarized in Table 1.

### 2.4. Sample Collection and Analysis

Surface water samples (1 L) were collected in amber glass bottles at four different points in the Quibú river basin, close to Habana City (Cuba). Two sampling points were close to a local hospital, whereas the other two sampling points were approximately 10 km far away from the hospital. Sodium azide was added to each sample after collection, and the samples were stored in the dark at 4 °C until analysis. All the samples were vacuum-filtered through a 0.22 µm nylon filter into glass vials to remove suspended matter. The samples (150 mL) were analysed following the optimized online MISPE–HPLC method and FQs concentrations were determined from the calibration curves. All the analyses were carried out in triplicate.

## 3. Results and Discussion

### 3.1. Online MISPE Optimization

In previous work, it was described [25] that the non-specific interactions between the FQs and the imprinted polymer can be minimized in the presence of mixtures of acetonitrile/water. FQs can be present in aqueous solutions as neutral, anionic and intermediate forms (zwitterions) due to the presence of carboxylic and amino groups. Therefore, their extraction behaviour will be pH-dependent. Thus, pH plays a key role in the retention of these compounds. In acidic conditions, functional monomers (pKa TFMA = 3.0, pKa MAA = 4.2) are protonated and their interaction with FQs is weak. On the other hand, MIP showed higher retention of the FQs in a pH range of 3–6, where the interaction MIP-FQ is consistent due to an ionic interaction between the protonated amino group of the FQs and the negatively charged functional monomers [21]. At higher pHs, no interaction was observed due to the deprotonation of the antimicrobials. Thus, a mixture of ACN/TFA 0.005% pH 3.0 (20:80, *v*/*v*) was selected as the washing solvent. Furthermore, due to the higher amount of mass of the polymer in the preconcentration column, in comparison to the conventional off-line MISPE cartridges used before, two additional steps with water, before and after the washing step, were introduced to equilibrate the preconcentration column. The recovery of the target FQs is depicted in Figure 3.

The recoveries obtained in these conditions were in the range 73.9–100.6% (RSD < 8.4%, *n* = 3) for the imprinted polymer and 0.3–4.2% (RSD < 6.9%, *n* = 3) for the nonimprinted material. Hence, these values confirm the presence of very high-affinity binding sites in the structure of the imprinted polymer, and the selected conditions minimize the retention resulting from the non-specific interactions. Thus, these conditions were selected for further experiments.

The effect of sample flow rate and volume was studied by a two-level factorial design and adding a centre point to the design, with the recovery of each FQ the target response. The design consisted of four experiments performed in triplicate and three replicates for the central point. In all the experiments, the amount for each FQ tested was fixed to 10 ng. The experimental domain and the results, in terms of average recovery, are shown in Table 2.

Recovery values were between 54 and 118% (RSD < 17%) for all the FQs tested. No significant differences in the recovery of the FQs were observed within the range of 0.5–2.5 mL min^−^^1^. On the other hand, the online system was able to percolate 200 mL of sample without significant loss of NOR, DANO, LOME, ENRO, or SARA at the three concentration levels tested (50, 91 and 500 ng mL^−^^1^ with recoveries of 78–118%, RSD < 17%), demonstrating the excellent capacity of the sorbent for these FQs. The differences in the recoveries obtained can be attributed to the different structure with the template molecule, specially in the part of the piperazinyl ring [22]. Nevertheless, in the experimental domain studied, LEVO and CIPRO showed slightly lower recoveries (47–82%, RSD < 6%). To achieve recovery of LEVO and CIPRO, the most common FQ used in Cuba, above 65%, flow rate and sample volume were optimized based on the results of the factorial design. Based on these results, a sample loading flow rate of 1.0 mL min^−^^1^ and a sample volume of 150 mL were selected for subsequent studies as a compromise between sample throughput and back pressure on the MISPE column. An example of a typical chromatogram obtained is shown in Figure 4.

### 3.2. Method Validation

The analytical method was validated in terms of linearity, precision, accuracy, sensitivity, limits of detection (LOD), and quantification (LOQ). Calibration solutions were prepared by spiking distilled water with each of the targeted FQs in the linear range of 0.5–100 μg for 150 mL volume, shown in Table 3. Good linearity was observed within the evaluated concentration ranges for all FQs tested (determination coefficient > 0.9843). The precision of this method was evaluated by measuring the RSDs of the inter-day tests. The experiments were carried out with the FQs spiked at different concentrations (0.7–666 ng L^−^^1^) in mineral water samples. The fortified samples were analysed on consecutive days, and all experiments were performed in triplicate. Inter-day recoveries (accuracy) were obtained from 65.7 to 100.6% with the RSDs (precision) less than 8.4%, indicating good within-laboratory precision achieved by the analytical procedure. Recoveries remained on a constant trend in the inter-day recovery curve, indicating satisfactory accuracy.

For comparison with other methods, LOD and LOQ were calculated based on the calibration curve, loading 150 mL of water for each calibration point (*n* = 3). The limit of detection (LOD) was calculated as 3.3S/slope and the limit of quantification (LOQ) as 10S/slope (S, standard deviation of the intercept). The LOD and LOQ were in the range of 0.1–0.7 ng L^−^^1^ and 0.4–1.5 ng L^−^^1^, respectively. From these results, it can be concluded that the online MISPE approach could reliably be used for the extraction of the target FQs in water samples. The use of a MIP as an online SPE sorbent minimizes sample treatment and the automatisation of the whole procedure is very efficient compared to previously reported methods (Table 4). Moreover, the method renders better LODs and LOQs in the determination of these antibiotics. LOD and LOQ were very similar to previous methods. However, most of the methods used for the determination of antibiotics are non-automated methods. More specifically, in the case of solid-phase extraction, all the methods developed with similar LODs up to now are off-line. The fact that the entire system is fully automated allows a decrease in the relative standard deviations (RSD%) in the determination of the antibiotics for these concentration levels. It also allows a considerable decrease in the analysis time per sample. Finally, the material has been able to be reused more than 50 times without losing its properties. This fact, combined with the small amount of polymer used in the preconcentration column (~50 mg), is translated into a significant decrease in the final cost of the analysis per sample.

### 3.3. Analysis of Quibú River Water

Analysis of the Quibú River water was carried out at four different points of its flow. Points A and B (Figure 5) correspond to a critical zone where effluents from a local hospital are discharged. The concentration of antibiotics in water at these points and the evolution of possible contaminants along the course of the river were studied. To this aim, the state of the waters was re-examined at distances of approximately 9 (point C) and 11 km (point D) from points A and B. As it can be observed in Table 5, at points A and B, the presence of the antibiotics NOR, ENRO and DANO could be observed in the Quibú river at ng L^−^^1^ concentration level, with the concentration of NOR being almost 10 times higher than other antibiotics. This fact makes sense since NOR is an antibiotic more widely used in different treatments than others that belong to this family. After several kilometres, at points C and D, the concentration of these analytes could not be quantified, so it is to be assumed that the concentration of antibiotics decreases along the course of the river, making impossible its quantification. In this way, it can be confirmed that the proposed method is valid for the punctual monitoring of water quality at ultratrace level for these antibiotics.

## 4. Conclusions

The present work has proven the application of a MIP as a robust sorbent for the simultaneous and automated online SPE determination of trace amounts of fluoroquinolones in water samples. The optimized procedure yields detection limits in the low ng L^−^^1^ levels in surface water with recoveries higher than 70% for all the target FQs. The MISPE column can be reused for more than 50 assays without losing its concentration efficiency. Finally, the developed method can be a useful tool for studying the occurrence and fate of FQs in the aquatic environment at a reasonable cost.

## Figures and Tables

**Figure 1 molecules-27-08120-f001:**
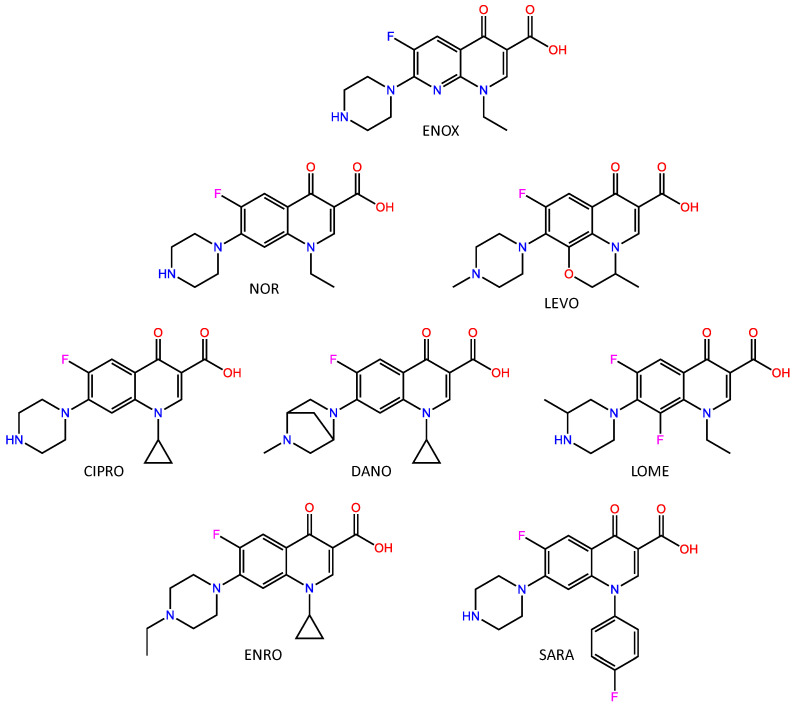
Chemical structures of the template molecule (ENOX) and the target FQs.

**Figure 2 molecules-27-08120-f002:**
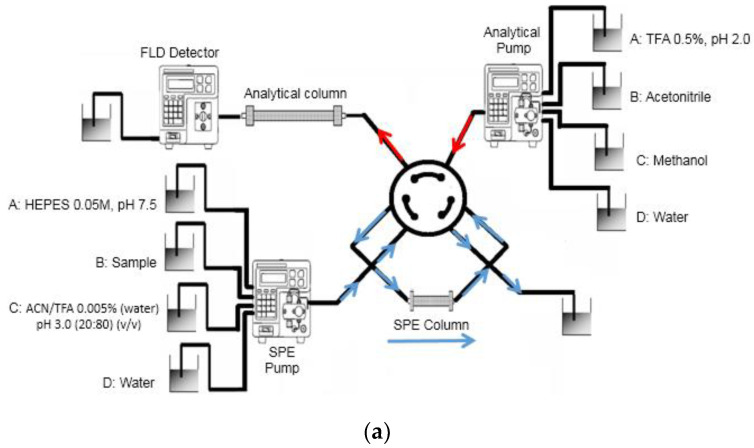
Scheme of the online MISPE-HPLC set-up. (**a**) Valve position A: sample loading; (**b**) valve position B: sample analysis.

**Figure 3 molecules-27-08120-f003:**
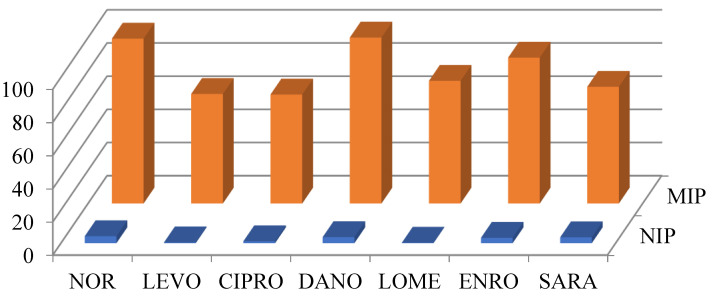
Recovery of target FQs in the MIP and NIP. FQs concentration: 500 ng L^−^^1^. Loading volume: 20 mL of water.

**Figure 4 molecules-27-08120-f004:**
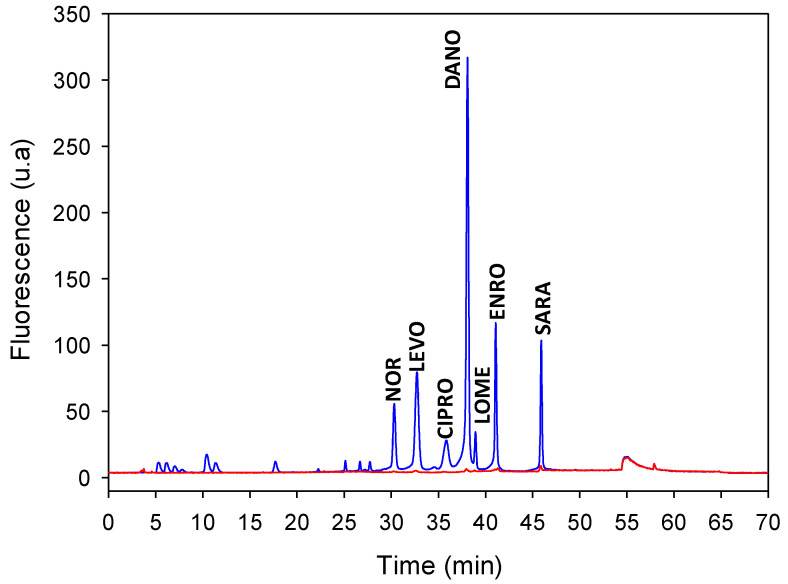
Chromatograms of target FQs in the MIP (blue line) and NIP (red line). FQs concentration: 67 ng L^−^^1^. Loading volume: 150 mL of water.

**Figure 5 molecules-27-08120-f005:**
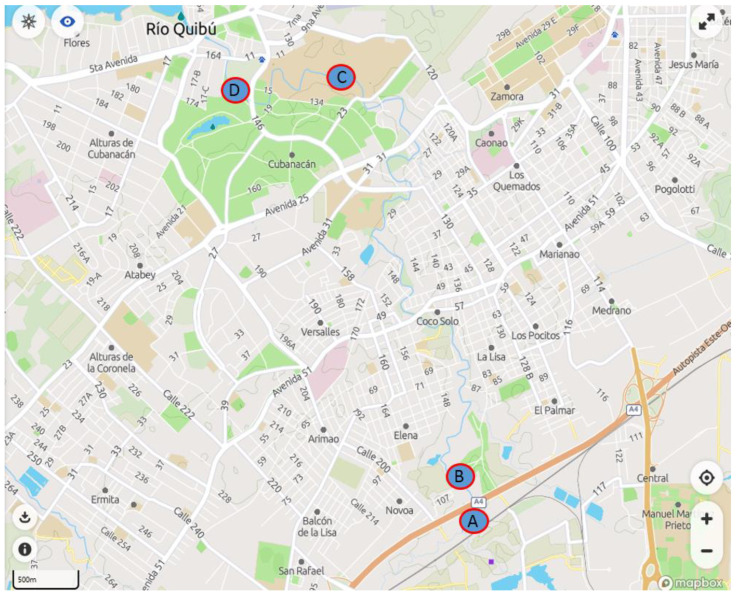
Surface water sampling points in the Quibú river basin.

**Table 1 molecules-27-08120-t001:** Preconcentration steps for the online MISPE analysis of water samples.

Step	Experimental Condition	V (mL)
1	Preconditioning: HEPES 0.05 M. pH 7.5	10
2	Loading of the sample in HEPES 0.05 M. pH 7.5	150
3	Washing 1: water	3
4	Washing 2: ACN/TFA 0.005% in water pH 3.0 (20:80)	3
5	Washing 3: water	3
6	Elution: ACN/TFA 0.5% in water. pH 2.0 (20:80)	10

**Table 2 molecules-27-08120-t002:** Experimental domain and two-level factorial design.

Parameter		Code	Level						
			Minimum			Central		Maximum	
Sample flow rate (mL min^−1^)		FR	0.5			1.5		2.5	
Sample volume (mL)		V	20			110		220	
**Experiment**	**V**	**FR**	**Recovery (%) ^1^**						
			**NOR**	**LEVO**	**CIPRO**	**DANO**	**LOME**	**ENRO**	**SARA**
1	20	0.5	96	61	82	112	83	99	89
2	20	2.5	96	54	75	116	80	90	94
3	110	1.5	97	47	61	112	90	98	84
4	220	0.5	105	78	77	105	118	92	93
5	220	2.5	98	59	67	101	100	70	78

^1^ Amount of FQ: 10 ng.

**Table 3 molecules-27-08120-t003:** Analytical characteristics of the online MISPE method.

FQ	Linear Range (ng L^−1^)	r^2^	LOD (ng L^−1^)	LOQ (ng L^−1^)	Recovery (%) ^1^
NOR	3.3–333	0.9974	0.3	0.9	99 (2.8)
LEVO	6.6–333	0.9977	0.3	1.0	66 (5.3)
CIPRO	6.6–333	0.9855	0.7	2.2	66 (8.4)
DANO	0.7–133	0.9843	0.3	0.7	101 (2.9)
LOME	6.6–666	0.9993	0.2	0.5	74 (3.8)
ENRO	3.3–333	0.9967	0.3	1.1	88 (2.8)
SARA	6.6–333	0.9995	0.1	0.4	70 (7.8)

^1^ FQ concentration: 67 ng L^−1^. RSD (%) in brackets (*n* = 3).

**Table 4 molecules-27-08120-t004:** Analytical methods for the determination of FQs in water samples.

Analytical Method	Analyte	LOD (ng L^−1^)	LOQ (ng L^−1^)	References
LC-MS/MS	NOR	36.3	121	[26]
	CIPRO	7.0	23.4	
	LOME	59	197	
	ENRO	55.1	184	
UA-IL-DLLME-LC-FLD	NOR	0.8	3	[27]
	CIPRO	4	13	
	DANO	0.8	3	
	LOME	13	43	
	ENRO	10	33	
SPE-LC-MS/MS	NOR	3.4	10.2	[15]
	CIPRO	3.3	10.1	
	ENRO	3.3	10.1	
SPE-LC-MS/MS	NOR	0.11	0.38	[28]
	CIPRO	0.09	0.29	
	ENRO	0.02	0.06	
MSPE-HPLC-MS/MS	NOR	1	-	[29]
	CIPRO	3	-	
	ENRO	3	-	
	SARA	2	-	
	LOME	3	-	
MSPE-HPLC-MS/MS	NOR	8.5	28	[30]
	CIPRO	23	78	
	ENRO	3.9	13	
	SARA	6	20	
	LOME	2.4	8.2	
	DANO	15	49	
SPE-HPLC/MS/MS	NOR	0.5	1.5	[31]
	CIPRO	1	3	
	ENRO	1	3	
	LOME	1	3	
	DANO	0.5	1.5	
MSPE/HPLC-DAD	CIPRO	20	-	[32]
	ENRO	20	-	
	DANO	10	-	
Electrochemical biosensor	CIPRO	29	-	[33]
Present work	NOR	0.3	0.9	
	LEVO	0.3	1.0	
	CIPRO	0.7	2.2	
	DANO	0.3	0.9	
	LOME	0.2	0.5	
	ENRO	0.3	1.1	
	SARA	0.1	0.4	

**Table 5 molecules-27-08120-t005:** Analysis of Quibú river water samples.

Sampling Point	FQ	Estimated Concentration (ng L^−1^)
A	NOR	21.3
	DANO	3.2
	LOME	3.5
B	NOR	21.7
	DANO	2.2
	LOME	2.3
C		Not found
D		Not found

RSD < 2% in all cases (*n* = 3).

## Data Availability

The data presented in this study are available on request from the corresponding authors.

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
