# Peer review of "The Ultratrace Determination of Fluoroquinolones in River Water Samples by an Online Solid-Phase Extraction Method Using a Molecularly Imprinted Polymer as a Selective Sorbent"

_molecules, 2022, doi:10.3390/molecules27238120_

Round 1
Reviewer 1 Report
The authors detected ultratrace fluoroquinolones in water samples by an on-line SPE-HPLC
-FLD method and a molecularly imprinted polymer as selective sorbent. This work was very interesting and obtained a very good result, which could be seen from the sensitive LOD and LOQ. But some errors should be revised. The manuscript also should be improved.
1: The topic is too wordy, especially about the analytical methods.
2: There are some errors in the abstract. In line 12, there are two “in” in the sentence.
In line 16, why “at the low ng/L”occurred in the sentence? Please check the sentence.
3: In Fig.1, why the template molecule not occurred?
4: In Fig 4, the two lines overlapped. Please revise the figure.
5: In Table 4, the linear range is somewhat strange, such as 333, 666, 3.3, 6.6. Why?
Furthermore, The LOQ 0.9 was higher than 0.7 in fifth line. In the last line of the table, 6.6 was much higher than the LOQ 0.4. Why ?
6: In Table 6, Not Detect, what is the meaning? The sample was not detected or the analytes were not found?
7: The English expression should be improved.
Reviewer 2 Report
Journal: Molecules
Ms. ID.: molecules-2026946
Title: Ultratrace determination of fluoroquinolones in river water samples by an on-line SPE-HPLC-FLD method and a Molecularly Imprinted Polymer as selective sorbent
Baeza et al. reported the development of an on-line solid phase extraction methodology based on a selective molecularly imprinted polymer (MIP) and fluorescent detection (HPLC-FLD) for the determination of fluoroquinolones (FQs) in water at the low ng L-1. The method was applied to determine FQs in river water. It is a very interesting manuscript. It fits well with the scope of the Journal. The results are promising. I consider the manuscript suitable for publication, but I also believe some important improvements are needed. The list of specific issues that should be addressed is listed below.
-There are many similar papers. The novelty of this manuscript should be emphasized.
-Table 1 should be moved to the Supplementary files.
-The results are very good and well-presented, but the discussion is thin. The authors should compare their results with other similar methods in the literature. Also, it would be useful to discuss their linearity range, LOD and LOQ with respect to the allowed quantities of the analyzed molecules. They should comment on the concentrations found in the real samples.
Author Response
Please see the attachmen

Round 2
Reviewer 2 Report
The authors improved the manuscript significantly. Still, they should emphasize the novelty of the paper in the Introduction.
Author Response
The authors improved the manuscript significantly. Still, they should emphasize the novelty of the paper in the Introduction.
A new paragraph in the introduction section has been added in order to improve the novelty.